# ADVERSARIAL PRIVACY PRESERVATION UNDER ATTRIBUTE INFERENCE ATTACK

## ABSTRACT

With the prevalence of machine learning services, crowdsourced data containing sensitive information poses substantial privacy challenges. Existing work focusing on protecting against membership inference attacks under the rigorous framework of differential privacy are vulnerable to attribute inference attacks. In light of the current gap between theory and practice, we develop a novel theoretical framework for privacy-preservation under the attack of attribute inference. Under our framework, we propose a minimax optimization formulation to protect the given attribute and analyze its privacy guarantees against arbitrary adversaries. On the other hand, it is clear that privacy constraint may cripple utility when the protected attribute is correlated with the target variable. To this end, we also prove an information-theoretic lower bound to precisely characterize the fundamental trade-off between utility and privacy. Empirically, we extensively conduct experiments to corroborate our privacy guarantee and validate the inherent trade-offs in different privacy preservation algorithms. Our experimental results indicate that the adversarial representation learning approaches achieve the best trade-off in terms of privacy preservation and utility maximization.

## 1 INTRODUCTION

With the growing demand for machine learning systems provided as services, a massive amount of data containing sensitive information, such as race, income level, age, etc., are generated and collected from local users. This poses a substantial privacy challenge and it has become an imperative object of study in machine learning (Abadi et al., 2016; Gilad-Bachrach et al., 2016), computer vision (Chou et al., 2018; Wu et al., 2018), healthcare (Beaulieu-Jones et al., 2018b;a), security (Shokri et al., 2017), and many other domains. In this paper, we consider a practical scenario where the prediction vendor requests crowdsourced data for a target task, e.g, scientific modeling. The data owner agrees on the data usage for the target task while she does not want her other private information (e.g., age, race) to be leaked. The goal of privacy-preserving in this context is then to protect private attributes of the sanitized data released by data owner from potential attribute inference attacks of a malicious adversary. For example, in an online advertising scenario, while the user (data owner) may agree to share her historical purchasing events, she also wants to protect her age information so that no malicious adversary can infer her age range from the shared data. Note that simply removing age attribute from the shared data is insufficient for this purpose, due to the redundant encoding in data, i.e., other attributes may have a high correlation with age.

Among many other techniques, differential privacy (DP) has been proposed and extensively investigated to protect the privacy of collected data (Dwork & Nissim, 2004; Dwork et al., 2006). DP embraces formal guarantees for privacy problems such as defending against the membership query attacks (Abadi et al., 2016; Papernot et al., 2016), or ensures the distribution of any two data records statistically indistinguishable (Erlingsson et al., 2014; Duchi et al., 2013; Bassily & Smith, 2015). However, DP still suffers from attribute inference attacks (Fredrikson et al., 2015; Cormode, 2011; Gong & Liu, 2016), as it only prevents an adversary from gaining additional knowledge by inclusion/exclusion of a subject, not from gaining knowledge from the data itself (Dwork et al., 2014). As a result, an adversary can still accurately infer sensitive attributes of data owners from differentially-private datasets. Such a gap between theory and practice calls for an important and appealing challenge:

*Can we find a representation of the raw data to remove private information related to a sensitive attribute while still preserving our utility of the target task? If no, what is the fundamental trade-off between privacy preservation and utility maximization?*

Clearly, under the setting of attribute inference attacks, the notion of privacy preservation should be attribute-specific: the goal is to protect specific attributes from being inferred by malicious adversaries. Note that this is in sharp contrast with differential privacy, where mechanisms are usually designed to resist worst-case membership query among all the data owners. From this perspective, our relaxed definition of privacy also allows for a more flexible design of algorithms with better utility.

**Our Contributions**   In this paper, we first formally define the notion of utility and privacy. We justify why our definitions are particularly suited under the setting of attribute inference attacks. Through the lens of representation learning, we then formulate the problem of utility maximization with privacy constraint as a minimax optimization problem that can be effectively and practically implemented. To provide a formal guarantee on privacy preservation, we prove an information-theoretic lower bound on the inference error of the protected attribute under attacks from arbitrary adversaries. To investigate the relationship between privacy preservation and utility maximization, we also provide a theoretical result to formally characterize the inherent trade-off between these two concepts. Empirically, we extensively conduct experiments to corroborate our privacy guarantee and validate the inherent trade-offs in different privacy preservation algorithms. From our empirical results, we conclude that the adversarial representation learning approach achieves the best trade-off in terms of privacy preservation and utility maximization, among various state-of-the-art privacy preservation algorithms.

## 2 PRELIMINARY

We first introduce our problem setting, the notations used throughout the paper and formally define the notions of utility and privacy discussed in this paper.

### 2.1 PROBLEM SETUP AND NOTATION

**Problem Setup**   We focus on the setting where the goal of the adversary is to perform attribute inference. This setting is ubiquitous in sever-client paradigm where machine learning is provided as a service (MLaaS, Ribeiro et al. (2015)). Formally, there are two parties in the system, namely the prediction vendor and the data owner. We consider the practical scenarios where users agree to contribute their data for training a machine learning model for specific purposes but do not want others to infer their private attributes in the data, such as health information, race, gender, etc. The prediction vendor will not collect raw user data but processed user data and the target attribute for the target task. In our setting, we assume the adversary cannot get other auxiliary information than the processed user data. In this case, the adversary can be anyone who can get access to the processed user data to some extent and wants to infer other private information. For example, malicious machine learning service providers are motivated to infer more information from users to do user profiling and targeted advertisements. The goal of the data owner is to provide as much information as possible to the prediction vendor to maximize the vendor's own utility, but under the constraint that the data owner should also protect the private information of the data source.

**Notation**   We use $\mathcal{X}$, $\mathcal{Y}$ and $\mathcal{A}$ to denote the input, output and adversary's output space, respectively. Accordingly, we use $X, Y, A$ to denote the random variables which take values in $\mathcal{X}, \mathcal{Y}$ and $\mathcal{A}$. We note that in our framework the input space $\mathcal{X}$ may or may not contain the private attribute $A$. For two random variables $X$ and $Y$, $I(X; Y)$ denotes the mutual information between $X$ and $Y$. We use $H(X)$ to mean the Shannon entropy of random variable $X$. Similarly, we use $H(X \mid Y)$ to denote the conditional entropy of $X$ given $Y$. We assume there is a joint distribution $\mathcal{D}$ over $\mathcal{X} \times \mathcal{Y} \times \mathcal{A}$ from which the data are sampled. To make our notation consistent, we use $\mathcal{D}_{\mathcal{X}}$, $\mathcal{D}_{\mathcal{Y}}$ and $\mathcal{D}_{\mathcal{A}}$ to denote the marginal distribution of $\mathcal{D}$ over $\mathcal{X}, \mathcal{Y}$ and $\mathcal{A}$. Given a feature map function $f : \mathcal{X} \to \mathcal{Z}$ that maps instances from the input space $\mathcal{X}$ to feature space $\mathcal{Z}$, we define $\mathcal{D}^f := \mathcal{D} \circ f^{-1}$ to be the induced (pushforward) distribution of $\mathcal{D}$ under $f$, i.e., for any event $E' \subseteq \mathcal{Z}$, $\Pr_{\mathcal{D}^f}(E') := \Pr_{\mathcal{D}}(f^{-1}(E')) = \Pr_{\mathcal{D}}(\{x \in \mathcal{X} \mid f(x) \in E'\})$.

## 2.2 Utility and Privacy

To simplify the exposition, we mainly discuss the attribute inference setting where $\mathcal{X} \subseteq \mathbb{R}^d$, $\mathcal{Y} = \mathcal{A} = \{0, 1\}$, but the underlying theory and methodology could easily be extended to the categorical case as well. In what follows, we shall first formally define both the *utility* of the prediction vendor and the *privacy* of the data owner. It is worth pointing out that our definition of privacy is *attribute-specific*, and this is in contrast with the classic framework of differential privacy where the goal is to preserve privacy in the general and worst-case query scenario. In particular, we seek to keep the utility of the data while being robust to an adversary on protecting specific information from attack.

A *hypothesis* is a function $h : \mathcal{X} \to \mathcal{Y}$. The *error* of a hypothesis $h$ under the distribution $\mathcal{D}$ over $\mathcal{X} \times \mathcal{Y}$ is defined as: $\mathrm{Err}(h) := \mathbb{E}_{\mathcal{D}}\big[|Y - h(X)|\big]$. Similarly, we use $\widehat{\mathrm{Err}}(h)$ to denote the empirical error of $h$ on a sample from $\mathcal{D}$. For binary classification problem, when $h(\mathbf{x}) \in \{0, 1\}$, the above loss also reduces to the error rate of classification. Let $\mathcal{H}$ be the Hilbert space of hypotheses. In the context of binary classification, we define the utility of a hypothesis $h \in \mathcal{H}$ as the opposite of error:

**Definition 2.1** (Utility). The utility of a hypothesis $h \in \mathcal{H}$ is $\mathrm{Util}(h) := 1 - \mathbb{E}_{\mathcal{D}}\big[|Y - h(X)|\big]$.

For binary classification, we always have $0 \leq \mathrm{Util}(h) \leq 1$, $\forall h \in \mathcal{H}$. Now we proceed to define a measure of *privacy* in our framework:

**Definition 2.2** (Privacy). The privacy w.r.t. task $A$ under attacks from $\mathcal{H}$ is defined as $\mathrm{Priv}_A(\mathcal{H}) := \min_{h \in \mathcal{H}} 1 - \big|\mathrm{Pr}_{\mathcal{D}}(h(X) = 1 \mid A = 1) - \mathrm{Pr}_{\mathcal{D}}(h(X) = 1 \mid A = 0)\big|$.

Again, it is straightforward to verify that $0 \leq \mathrm{Priv}_A(\mathcal{H}) \leq 1$. Based on our definition, $\mathrm{Priv}_A(\mathcal{H})$ then measures the privacy of data under possible attacks from adversaries in $\mathcal{H}$. We can also refine the above definition to a particular hypothesis $h : \mathcal{X} \to \{0, 1\}$ to measure its ability to steal information about $A$: $\mathrm{Priv}_A(h) = 1 - \big|\mathrm{Pr}_{\mathcal{D}}(h(X) = 1 \mid A = 1) - \mathrm{Pr}_{\mathcal{D}}(h(X) = 1 \mid A = 0)\big|$.

**Proposition 2.1.** Let $h : \mathcal{X} \to \{0, 1\}$ be a hypothesis, then $\mathrm{Priv}_A(h) = 1$ iff $I(h(X); A) = 0$ and $\mathrm{Priv}_A(h) = 0$ iff $h(X) = A$ almost surely or $h(X) = 1 - A$ almost surely.

Proposition 2.1 justifies our definition of privacy: when $\mathrm{Priv}_A(h) = 1$, it means that $h(X)$ contains no information about the sensitive attribute $A$. On the other hand, if $\mathrm{Priv}_A(h) = 0$, then $h(X)$ fully predicts $A$ (or equivalently, $1 - A$) from input $X$. In the latter case $h(X)$ also contains perfect information of $A$ in the sense that $I(h(X); A) = H(A)$, i.e., the Shannon entropy of $A$. It is worth pointing out that our definition of privacy is insensitive to the marginal distribution of $A$, and hence is more robust than other definitions such as the error rate of predicting $A$. In that case, if $A$ is extremely imbalanced, even a naive predictor can attain small prediction error by simply outputting constant. We call a hypothesis space $\mathcal{H}$ *symmetric* if $\forall h \in \mathcal{H}$, $1 - h \in \mathcal{H}$ as well. Interestingly, when $\mathcal{H}$ is symmetric, we can also relate the privacy $\mathrm{Priv}_A(\mathcal{H})$ to a binary classification problem:

**Proposition 2.2.** If $\mathcal{H}$ is symmetric, then $\mathrm{Priv}_A(\mathcal{H}) = \min_{h \in \mathcal{H}} \mathrm{Pr}(h(X) = 0 \mid A = 1) + \mathrm{Pr}(h(X) = 1 \mid A = 0)$.

**Remark** Consider the following confusion matrix between the actual private attribute $A$ and its predicted variable $h(X)$ in Table 1. The false positive rate (eqv. Type-I error) is defined as FPR = FP / (FP + TN) and the false negative rate (eqv. Type-II error) is similarly defined as FNR = FN / (FN + TP). Using the terminology of confusion matrix, it is then clear that $\mathrm{Pr}(h(X) = 0 \mid A = 1) = \mathrm{FNR}$ and $\mathrm{Pr}(h(X) = 1 \mid A = 0) = \mathrm{FPR}$. In other words, Proposition 2.2 says that if $\mathcal{H}$ is symmetric, then the privacy of a hypothesis space $\mathcal{H}$ corresponds to the minimum sum of Type-I and Type-II error that is achievable under attacks from $\mathcal{H}$.

Table 1: Confusion matrix between $A$ and $h(X)$.

|           | $h(X) = 0$ | $h(X) = 1$ |
|-----------|------------|------------|
| $A = 0$   | TN         | FP         |
| $A = 1$   | FN         | TP         |

## 3 Minimax Optimization against Attribute Inference Attacks

### 3.1 Minimax Formulation

Given a set of samples $\mathbf{S} = \{(\mathbf{x}_i, y_i, a_i)\}_{i=1}^n$ drawn i.i.d. from the joint distribution $\mathcal{D}$, how can the data owner keeps the utility of the data while keeping the sensitive attribute $A$ private under potential

attacks from malicious adversary? Through the lens of representation learning, we seek to find a (non-linear) feature representation $f : \mathcal{X} \to \mathcal{Z}$ from input space $\mathcal{X}$ to feature space $\mathcal{Z}$ such that $f$ still preserves relevant information w.r.t. the target task of inferring $Y$ while hiding sensitive attribute $A$. Specifically, we can solve the following unconstrained regularized problem with $\lambda > 0$:

$$\min_{h \in \mathcal{H}, f} \max_{h' \in \mathcal{H}} \widehat{\mathrm{Err}}(h \circ f) - \lambda \big( \Pr_{\mathbf{S}}(h'(f(X)) = 0 \mid A = 1) + \Pr_{\mathbf{S}}(h'(f(X)) = 1 \mid A = 0) \big) \quad (1)$$

It is worth pointing out that the optimization formulation in (1) admits an interesting game-theoretic interpretation, where two agents $f$ and $h'$ play a game whose score is defined by the objective function in (1). Intuitively, $h'$ seeks to minimize the sum of Type-I and Type-II error while $f$ plays against $h'$ by learning transformation to removing information about the sensitive attribute $A$. Algorithmically, for the data owner to achieve the goal of hiding information about the sensitive attribute $A$ from malicious adversary, it suffices to learn representation that is independent of $A$. Formally:

**Proposition 3.1.** Let $f : \mathcal{X} \to \mathcal{Z}$ be a deterministic function and $\mathcal{H} \subseteq 2^{\mathcal{Z}}$ be a hypothesis class over $\mathcal{Z}$. For any joint distribution $\mathcal{D}$ over $X, A, Y$, if $I(f(X); A) = 0$, then $\mathrm{Priv}_A(\mathcal{H} \circ f) = 1$.

Note that in this sequential game, $f$ is the first-mover and $h'$ is the second. Hence without explicit constraint $f$ possesses a first-mover advantage so that $f$ can dominate the game by simply mapping all the input $X$ to a constant or uniformly random noise. To avoid these degenerate cases, the first term in the objective function of (1) acts as an incentive to encourage $f$ to preserve task-related information. But will this incentive compromise our privacy? As an extreme case if the target variable $Y$ and the sensitive attribute $A$ are perfectly correlated, then it should be clear that there is a trade-off in achieving utility and preserving privacy. In Sec. 4 we shall provide an information-theoretic bound to precisely characterize such inherent trade-off. Furthermore, although the formulation in (1) only works for defense over a single attribute, it is straightforward to extend the formulation so that it can protect attacks over multiple attributes. Due to space limit, we defer the discussion of this extension to Section C in appendix.

## 3.2 Privacy Guarantees on Attribute Inference Attacks

In the last section we propose the unconstrained minimax formulation (1) to optimize both our utility and the defined privacy measure. Clearly, the hyperparameter $\lambda$ measures the trade-off between utility and our privacy. On one hand, if $\lambda \to 0$, we barely care about the privacy and devote all the focus to maximize our utility. On the other extreme, if $\lambda \to \infty$, we are only interested in protecting the privacy. In what follows we analyze the true error that an optimal adversary has to incur in the limit when both the task classifier and the adversary have unlimited capacity, i.e., they can be any randomized functions from $\mathcal{Z}$ to $\{0, 1\}$. To study the true error, we hence use the population loss rather than the empirical loss in our objective function. Furthermore, since the binary classification error in (1) is NP-hard to optimize even for hypothesis class of linear predictors, in practice we consider the cross-entropy loss function as a convex surrogate loss. The cross-entropy loss $\mathrm{CE}_Y(h)$ of a probabilistic hypothesis $h : \mathcal{X} \to [0, 1]$ w.r.t. $Y$ on a distribution $\mathcal{D}$ is defined as follows:

$$\mathrm{CE}_Y(h) := -\mathbb{E}_{\mathcal{D}} \left[ \mathbb{I}(Y = 0) \log(1 - h(X)) + \mathbb{I}(Y = 1) \log(h(X)) \right]. \quad (2)$$

With a slight abuse of notation, we use $\mathrm{CE}_A(h')$ to mean the cross-entropy loss of the adversary $h'$ w.r.t. $A$. Using the same notation, the optimization formulation with cross-entropy loss becomes:

$$\min_{h \in \mathcal{H}, f} \max_{h' \in \mathcal{H}} \mathrm{CE}_Y(h \circ f) - \lambda \mathrm{CE}_A(h' \circ f) \quad (3)$$

Given a feature map $f : \mathcal{X} \to \mathcal{Z}$, assume that $\mathcal{H}$ contains all the possible randomized classifiers from the feature space $\mathcal{Z}$ to $\{0, 1\}$. For example, a randomized classifier can be constructed by first defining a probabilistic function $h : \mathcal{Z} \to [0, 1]$ followed by a random coin flipping to determine the output label, where the probability of the coin being 1 is given by $h(Z)$. Under such assumptions, the following lemma shows that the optimal target classifier under $f$ is given by the conditional distribution $h^*(Z) := \Pr(Y = 1 \mid Z)$.

**Lemma 3.1.** For any feature map $f : \mathcal{X} \to \mathcal{Z}$, assume that $\mathcal{H}$ contains all the randomized binary classifiers, then o and $h^*(Z) := \arg\min_{h \in \mathcal{H}} \mathrm{CE}_Y(h \circ f) = \Pr(Y = 1 \mid Z = f(X))$.

By a symmetric argument, we can also see that the worst-case (optimal) adversary under $f$ is the conditional distribution $h^{'*}(Z) := \Pr(A = 1 \mid Z)$ and $\min_{h' \in \mathcal{H}} \mathrm{CE}_A(h' \circ f) = H(A \mid Z)$. Hence

we can further simplify the optimization formulation (3) to the following form where the only optimization variable is the feature map $f$:

$$\min_f \quad H(Y \mid Z = f(X)) - \lambda H(A \mid Z = f(X)) \tag{4}$$

Since $Z = f(X)$ is a deterministic feature map, it follows from the basic properties of Shannon entropy that

$$H(Y \mid X) \leq H(Y \mid Z = f(X)) \leq H(Y), \quad H(A \mid X) \leq H(A \mid Z = f(X)) \leq H(A),$$

which means that $H(Y \mid X) - \lambda H(A)$ is a lower bound of the optimum of the objective function in (4). However, such lower bound is not necessarily achievable. To see this, consider the simple case where $Y = A$ almost surely. In this case there exists no deterministic feature map $Z = f(X)$ that is both a sufficient statistics of $X$ w.r.t. $Y$ while simultaneously filters out all the information w.r.t. $A$ except in the degenerate case where $A(Y)$ is constant. On the other hand, to show that solving the optimization problem in (4) helps to protect our privacy, the following theorem gives a bound of privacy in terms of the error that has to be incurred by the optimal adversary:

**Theorem 3.1.** Let $f^*$ be the optimal feature map of (4) and define $H^* := H(A \mid Z = f^*(X))$. Then for any adversary $\widehat{A} : \mathcal{Z} \to \{0, 1\}$ such that $\widehat{A} \perp A \mid Z$, $\Pr_{\mathcal{D}^{f^*}}(\widehat{A} \neq A) \geq H^*/2 \lg(6/H^*)$.

**Remark**  Theorem 3.1 shows that whenever the conditional entropy $H^* = H(A \mid Z = f^*(X))$ is large, then the inference error of the protected attribute incurred by any (randomized) adversary has to be at least $\Omega(H^*/\log(1/H^*))$. As we have already shown above, the conditional entropy essentially corresponds to the second term in our objective function, whose optimal value could further be flexibly adjusted by tuning the trade-off parameter $\lambda$. As a final note, Theorem 3.1 also shows that representation learning helps to protect the privacy about $A$ since we always have $H(A \mid Z = f(X)) \geq H(A \mid X)$ for any deterministic feature map $f$ so that the lower bound of inference error by any adversary is larger after learning the representation $Z = f(X)$.

## 4  Inherent trade-off between Utility and Privacy

As we briefly mentioned in Sec. 3.1, when the protected sensitive attribute $A$ and the target variable $Y$ are perfectly correlated, it is impossible to simultaneously achieve the goal of privacy-preserving and utility-maximizing. But what is the exact trade-off between utility and privacy when they are correlated? In this section we shall provide an information-theoretic bound to quantitatively characterize the inherent trade-off between privacy and utility, due to the discrepancy between the conditional distributions of the target variable given the sensitive attribute. Our result is algorithm-independent, hence it applies to a general setting where there is a need to preserve both utility and privacy. To the best of our knowledge, this is the first information-theoretic result to precisely quantify such trade-off. Due to space limit, we defer all the proofs to appendix.

Before we proceed, we first define several information-theoretic concepts that will be used in our analysis. For two distributions $\mathcal{D}$ and $\mathcal{D}'$, the Jensen-Shannon (JS) divergence $D_{\text{JS}}(\mathcal{D}, \mathcal{D}')$ is:

$$D_{\text{JS}}(\mathcal{D}, \mathcal{D}') := \frac{1}{2} D_{\text{KL}}(\mathcal{D} \mid\mid \mathcal{D}_M) + \frac{1}{2} D_{\text{KL}}(\mathcal{D}' \mid\mid \mathcal{D}_M),$$

where $D_{\text{KL}}(\cdot \mid\mid \cdot)$ is the Kullback–Leibler (KL) divergence and $\mathcal{D}_M := (\mathcal{D} + \mathcal{D}')/2$. The JS divergence can be viewed as a symmetrized and smoothed version of the KL divergence, and it is upper bounded by the $L_1$ distance (total variation) between two distributions through Lin's Lemma:

**Lemma 4.1** (Lin (1991)). Let $\mathcal{D}$ and $\mathcal{D}'$ be two distributions, then $D_{\text{JS}}(\mathcal{D}, \mathcal{D}') \leq \frac{1}{2} ||\mathcal{D} - \mathcal{D}'||_1$.

Unlike the KL divergence, the JS divergence is bounded: $0 \leq D_{\text{JS}}(\mathcal{D}, \mathcal{D}') \leq 1$. Additionally, from the JS divergence, we can define a distance metric between two distributions as well, known as the JS distance (Endres & Schindelin, 2003): $d_{\text{JS}}(\mathcal{D}, \mathcal{D}') := \sqrt{D_{\text{JS}}(\mathcal{D}, \mathcal{D}')}$. With respect to the JS distance, for any feature space $\mathcal{Z}$ and any deterministic mapping $f : \mathcal{X} \to \mathcal{Z}$, we can prove the following lemma via the celebrated data processing inequality:

**Lemma 4.2.** Let $\mathcal{D}_0$ and $\mathcal{D}_1$ be two distributions over $\mathcal{X}$ and let $\mathcal{D}_0^f$ and $\mathcal{D}_1^f$ be the induced distributions of $\mathcal{D}_0$ and $\mathcal{D}_1$ over $\mathcal{Z}$ by function $f$, then $d_{\text{JS}}(\mathcal{D}_0^f, \mathcal{D}_1^f) \leq d_{\text{JS}}(\mathcal{D}_0, \mathcal{D}_1)$.

Without loss of generality, any method aiming to predict the target variable $Y$ defines a Markov chain as $X \xrightarrow{f} Z \xrightarrow{h} \hat{Y}$, where $\hat{Y}$ is the predicted target variable given by hypothesis $h$ and $Z$ is the intermediate representation defined by the feature mapping $f$. Hence for any distribution $\mathcal{D}_0(\mathcal{D}_1)$ of $X$, this Markov chain also induces a distribution $\mathcal{D}_0^{h \circ f}(\mathcal{D}_1^{h \circ f})$ of $\hat{Y}$ and a distribution $\mathcal{D}_0^f(\mathcal{D}_1^f)$ of $Z$. Now let $\mathcal{D}_0^Y(\mathcal{D}_1^Y)$ be the underlying true conditional distribution of $Y$ given $A = 0(A = 1)$. Realize that the JS distance is a metric, the following chain of triangular inequalities holds:

$$d_{\mathrm{JS}}(\mathcal{D}_0^Y, \mathcal{D}_1^Y) \leq d_{\mathrm{JS}}(\mathcal{D}_0^Y, \mathcal{D}_0^{h \circ f}) + d_{\mathrm{JS}}(\mathcal{D}_0^{h \circ f}, \mathcal{D}_1^{h \circ f}) + d_{\mathrm{JS}}(\mathcal{D}_1^{h \circ f}, \mathcal{D}_1^Y).$$

Combining the above inequality with Lemma 4.2 to show

$$d_{\mathrm{JS}}(\mathcal{D}_0^{h \circ f}, \mathcal{D}_1^{h \circ f}) \leq d_{\mathrm{JS}}(\mathcal{D}_0^f, \mathcal{D}_1^f),$$

we immediately have:

$$d_{\mathrm{JS}}(\mathcal{D}_0^Y, \mathcal{D}_1^Y) \leq d_{\mathrm{JS}}(\mathcal{D}_0^Y, \mathcal{D}_0^{h \circ f}) + d_{\mathrm{JS}}(\mathcal{D}_0^f, \mathcal{D}_1^f) + d_{\mathrm{JS}}(\mathcal{D}_1^{h \circ f}, \mathcal{D}_1^Y).$$

Intuitively, $d_{\mathrm{JS}}(\mathcal{D}_0^Y, \mathcal{D}_0^{h \circ f})$ and $d_{\mathrm{JS}}(\mathcal{D}_1^Y, \mathcal{D}_1^{h \circ f})$ measure the distance between the predicted and the true target distribution on $A = 0/1$ cases, respectively. Formally, let $\mathrm{Err}_a(h \circ f)$ be the prediction error of function $h \circ f$ conditioned on $A = a$. With the help of Lemma 4.1, the following result establishes a relationship between $d_{\mathrm{JS}}(\mathcal{D}_a^Y, \mathcal{D}_a^{h \circ f})$ and the utility of $h \circ f$:

**Lemma 4.3.** Let $\hat{Y} = h(f(X)) \in \{0, 1\}$ be the predictor, then for $a \in \{0, 1\}$, $d_{\mathrm{JS}}(\mathcal{D}_a^Y, \mathcal{D}_a^{h \circ f}) \leq \sqrt{\mathrm{Err}_a(h \circ f)}$.

Combine Lemma 4.2 and Lemma 4.3, we get the following key lemma that is the backbone for proving the main results in this section:

**Lemma 4.4** (Key lemma). Let $\mathcal{D}_0, \mathcal{D}_1$ be two distributions over $\mathcal{X} \times \mathcal{Y}$ conditioned on $A = 0$ and $A = 1$ respectively. Assume the Markov chain $X \xrightarrow{f} Z \xrightarrow{h} \hat{Y}$ holds, then $\forall h \in \mathcal{H}$:

$$d_{\mathrm{JS}}(\mathcal{D}_0^Y, \mathcal{D}_1^Y) \leq \sqrt{\mathrm{Err}_0(h \circ f)} + d_{\mathrm{JS}}(\mathcal{D}_0^f, \mathcal{D}_1^f) + \sqrt{\mathrm{Err}_1(h \circ f)}. \tag{5}$$

We emphasize that for $a \in \{0, 1\}$, the term $\mathrm{Err}_a(h \circ f)$ measures the conditional error of the predicted variable $\hat{Y}$ by composite function $h \circ f$ over $\mathcal{D}_a$. Similarly, we can define the *conditional utility* for $a \in \{0, 1\} : \mathrm{Util}_a(h \circ f) := 1 - \mathrm{Err}_a(h \circ f)$. The following main theorem then characterizes a fundamental trade-off between utility and privacy:

**Theorem 4.1.** Let $\mathcal{H} \subseteq 2^{\mathcal{Z}}$ contains all the classifiers from $\mathcal{Z}$ to $\{0, 1\}$. Given the conditions in Lemma 4.4, $\forall h \in \mathcal{H}$, $\mathrm{Util}_0(h \circ f) + \mathrm{Util}_1(h \circ f) + \mathrm{Priv}_A(\mathcal{H} \circ f) \leq 3 - \frac{1}{3} D_{\mathrm{JS}}(\mathcal{D}_0^Y, \mathcal{D}_1^Y)$.

A few remarks follow. First, note that the maximal value achievable by the sum of the three terms on the L.H.S. is 3. In light of this, the upper bound given in Theorem 4.1 shows that when the marginal distribution of the target variable $Y$ differ between two cases $A = 0$ or $A = 1$, then it is impossible to perfectly maximize utility and privacy. Furthermore, the trade-off due to the difference in marginal distributions is precisely given by the JS divergence $D_{\mathrm{JS}}(\mathcal{D}_0^Y, \mathcal{D}_1^Y)$. Note that in Theorem 4.1 the upper bound holds for *any* hypothesis $h$ in the richest hypothesis class $\mathcal{H}$ that contains all the possible binary classifiers. Put it another way, if we would like to maximally preserve privacy w.r.t. sensitive attribute $A$, then we have to incur a large joint error:

**Theorem 4.2.** Assume the conditions in Theorem 4.1 hold. If $\mathrm{Priv}_A(\mathcal{H} \circ f) \geq 1 - D_{\mathrm{JS}}(\mathcal{D}_0^Y, \mathcal{D}_1^Y)$, then $\forall h \in \mathcal{H}$, $\mathrm{Err}_0(h \circ f) + \mathrm{Err}_1(h \circ f) \geq \frac{1}{2}\left(d_{\mathrm{JS}}(\mathcal{D}_0^Y, \mathcal{D}_1^Y) - \sqrt{1 - \mathrm{Priv}_A(\mathcal{H} \circ f)}\right)^2$.

**Remark** The above lower bound characterizes a fundamental trade-off between privacy and joint error. In particular, up to a certain level $1 - D_{\mathrm{JS}}(\mathcal{D}_0^Y, \mathcal{D}_1^Y)$, the larger the privacy, the larger the joint error. In light of Proposition 3.1, this means that although the data-owner, or the first-mover $f$, could try to maximally preserve the privacy via constructing $f$ such that $f(X)$ is independent of $A$, such construction will also inevitably compromise the joint utility of the prediction vendor. It is also worth pointing out that our results in both Theorem 4.1 and Theorem 4.2 are attribute-independent in the sense that neither of the bounds depends on the marginal distribution of $A$. Instead, all the terms in our results only depend on the conditional distributions given $A = 0$ and $A = 1$. This is often more desirable than bounds involving mutual information, e.g., $I(A, Y)$, since $I(A, Y)$ is close to 0 if the marginal distribution of $A$ is highly imbalanced.

## 5 EXPERIMENTS

Our theoretical results on the privacy guarantee of attribute inference attacks imply that the inference error of the protected attribute incurred by any (randomized) adversary has to be at least $\Omega(H^*/\log(1/H^*))$. In this section we extensively conduct experiments on two real-world benchmark datasets, the UCI Adult dataset (Dua & Graff, 2017) and the UTKFace dataset (Zhang et al., 2017) to verify 1). Our guarantee on privacy can be used as a certificate for different privacy preservation methods. 2). Inherent trade-offs exist between privacy and utility exist in all methods. 3). Among all the privacy preservation algorithms, including differential privacy, the adversarial representation learning approach achieves the best trade-off in terms of privacy preservation and utility maximization.

### 5.1 DATASETS AND SETUP

**Datasets**   1). Adult dataset: The Adult dataset is a benchmark dataset for privacy-preservation. The task is to predict whether an individual's income is greater or less than 50K/year based on census data. The attributes in the dataset includes gender, education, occupation, age, etc. In this experiment we set the target task as income prediction and the private task as inferring gender, age and education, respectively. 2). The UTKFace dataset is a large-scale face dataset containing more than 20,000 images with annotations of age, gender, and ethnicity. It is one of the benchmark datasets for age estimation, gender and race classifications. In this experiment, we set our target task as gender classification and we use the age and ethnicity as the protected attributes. We refer readers to Section D in the appendix for detailed descriptions about the data pre-processing pipeline.

**Methods**   We conduct extensive experiments with the following methods to verify our theoretical results and provide a thorough practical comparison among these methods. 1). Privacy Partial Least Squares (PPLS) (Enev et al., 2012), 2). Privacy Linear Discriminant Analysis (PLDA) (Whitehill & Movellan, 2012), 3). Minimax filter with alternative update (ALT-UP) (Hamm, 2017), 4) Maximum Entropy Adversarial Representation Learning (MAX-ENT) (Roy & Boddeti, 2019) 5). Gradient Reversal Layer (GRL) (Ganin et al., 2016) 6). Principal Component Analysis (PCA) 7). No defense (NO-DEF), 8) Local Differential Privacy (LDP) with Laplacian mechanism, 9). differentially private SGD (DPSGD) (Abadi et al., 2016).

Among the first seven methods, the first five are state-of-the-art minimax methods for protecting against attribute inference attacks while the latter two are non-private baselines for comprehensive comparison. Although DP is not tailored to attribute inference attack, we can still add two DP baselines to examine the utility and privacy trade-off for comparison. Our goal here is to provide a thorough comparison in terms of utility-privacy trade-off by using methods from both representation learning and differential privacy.

To make sure the comparison is fair among different methods, we conduct a controlled experiment by using the same network structure as the baseline hypothesis among all the methods for each dataset. For each experiment on the Adult dataset and UTKFace dataset, we repeat the experiments for 10 times to report both the average performance and their standard deviations. Details on network structures and implementation of the above algorithms are provided in the appendix.

Note that in practice due to the non-convexity nature of optimizing deep neural nets, we cannot guarantee to find the global optimal conditional entropy $H^*$. Hence in order to compute the privacy guarantee given by our lower bound in Theorem 3.1, we use the cross-entropy loss of the optimal adversary found by our algorithm on inferring the sensitive attribute $A$. Furthermore, since our analysis only applies to representation learning based approaches, we do not have a privacy guarantee for DP-related methods in our context.

### 5.2 RESULTS AND ANALYSIS

We visualize the performances of the aforementioned algorithms on privacy preservation and utility maximization in Figure 1 and Figure 2, respectively. First, from Figure 1, we can see that among all the methods, both LDP, PLDA, ALT-UP, MAX-ENT and GRL are effective in privacy preservation by forcing the optimal adversary to incur a large inference error. On the other hand, PCA and NO-DEF are the least effective ones. This is expected as either NO-DEF nor PCA tries to filter information in

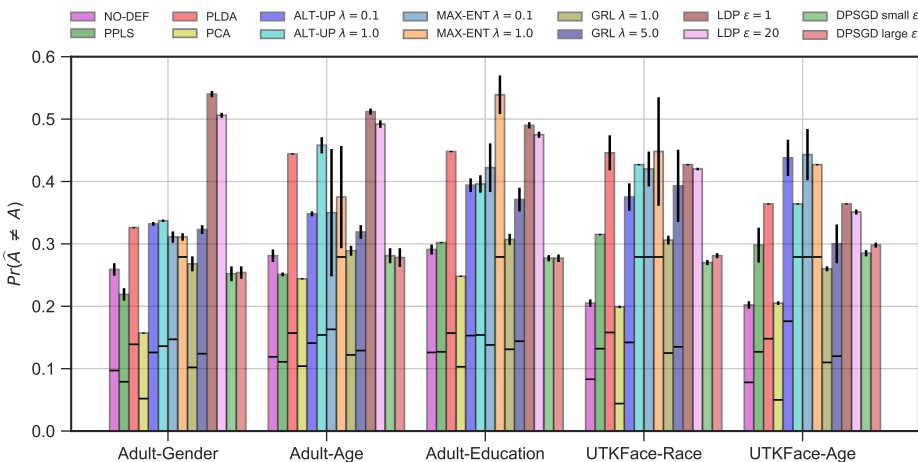

Figure 1: Performance on privacy preservation of different methods (the larger the better). The horizontal lines across the bars indicate the corresponding privacy guarantees given by our lower bound in Theorem 3.1.

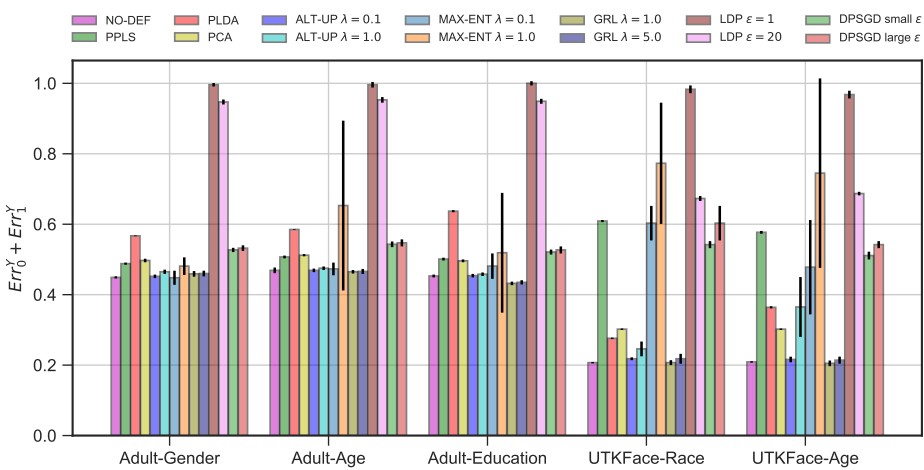

Figure 2: The joint conditional error ($\mathrm{Err}_0 + \mathrm{Err}_1$, the smaller the better) of different methods.

data about the sensitive attribute $A$. We can also see that with a larger trade-off value $\lambda$, ALT-UP, MAX-ENT and GRL achieve better privacy preservation.

Second, from Figure 2, we can also see a sharp contrast between DP-based methods and other methods in terms of the joint conditional error on the target task: both LDP and DPSGD incur significant utility loss compared with other methods. Combining this one with our previous observation from Figure 1, we can see that DP either makes data private by adding large amount of noise to effectively filter out all the information available in the data, including both target-related and sensitive information, or add insufficient amount of noise so that both target-related and sensitive information is well preserved. As a comparison, representation learning based approaches leads to a much better trade-off. Among all the representation learning methods, PLDA, ALT-UP, MAX-ENT and GRL perform the best in privacy preservation. Compared to ALT-UP and GRL, MAX-ENT and PLDA is more effective in privacy preservation in some cases, but at the cost of a significant drop in utility. It is also worth to note that different adversarial representation learning methods have different sensitivity on $\lambda$: a large $\lambda$ for MAX-ENT can lead to an unstable model training process and result in a large utility loss. In contrast, GRL is often more stable, which is consistent to the results shown in (Daskalakis & Panageas, 2018).

## 6 RELATED WORK

The attribute inference attack problem has close connections to both differential privacy and algorithmic fairness. In this section we mainly focus on discussing the connections and differences between these problems. As a summary, we visualize their relationships in the diagram shown in Figure 3.

**Differential Privacy** DP has been proposed to bound the difference of algorithmic output between any two "neighboring" datasets from the released data (Dwork & Nissim, 2004; Dwork et al., 2006; Erlingsson et al., 2014) and was used in the training of deep neural network recently (Abadi et al., 2016; Papernot et al., 2016; Phan et al., 2017). Our definition of privacy is different from (local) differential privacy since the goal of DP tries to make any two neighboring datasets have close probabilities to produce the same output. In the setting of learning algorithms, this means that the models trained from two neighboring datasets should be close to each other. However, this does not necessarily imply that the learned model itself is free from attribute inference attacks. As a comparison, our goal of defending attribute inference attacks is to learn a representation such that the protected attributes cannot be accurately inferred. Put it in another way, given a dataset matrix, the goal of DP is to ensure that it is hard to infer about a row in the matrix while our privacy definition seeks to ensure that it is hard to infer about a specific column of the data matrix. From this perspective, DP is closely related to the well-known membership inference attack (Shokri et al., 2017) instead. It is observed (Dwork et al., 2012) that the notion of individual fairness may be viewed as a generalization of DP.

**Algorithmic Fairness** The privacy defined in this work is related to the notion of group fairness in the literature of algorithmic fairness (Dwork et al., 2012; Edwards & Storkey, 2015). In particular, adversarial learning methods have been used as a tool in both fields to achieve the corresponding goals. However, the motivations and goals significantly differ between these two fields. Specifically, the widely adopted notion of group fairness, namely equalized odds (Hardt et al., 2016), requires equalized false positive and false negative rates across different demographic subgroups. As a comparison, in applications where privacy is a concern, we mainly want to ensure that adversaries cannot steal sensitive information from the data. Hence our goal is to give a worst case guarantee on the inference error that any adversary has at least to incur. To the best of our knowledge, our results in Theorem 3.1 is the first one to analyze the performance of privacy preservation in such scenarios. Furthermore, no prior theoretical results exist on discussing the trade-off between privacy and utility on defending attribute inference attacks. Our proof techniques developed in this work could also be used to derive information-theoretic lower bounds in related problems as well (Zhao et al., 2019; Zhao & Gordon, 2019).

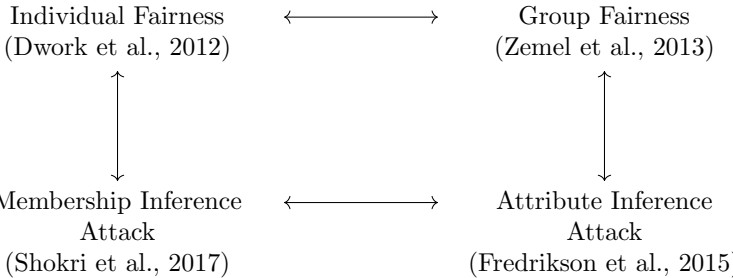

Figure 3: Relationships between different notions of fairness and privacy.

## 7 CONCLUSION

We develop a theoretical framework for privacy preservation under the setting of attribute inference attacks. Under this setting, we propose a theoretical framework that suggests using adversarial learning techniques to protect the private attribute. We further analyze the privacy guarantee of the defense method in the limit of worst-case adversaries and prove an information-theoretic lower bound to quantify the inherent trade-off between utility and privacy. Following our formulation, we conduct

extensive experiments to corroborate our theoretical results and to empirically compare different state-of-the-art privacy preservation algorithms. Experimental results show that the adversarial representation learning approaches are very effective in defending attribute inference attacks and often achieve the best trade-off in terms of privacy preservation and utility maximization. We believe our work takes an important step towards better understanding the privacy-utility trade-off, and it also helps to stimulate the future design of privacy-preservation algorithm with adversarial learning techniques.

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

## APPENDIX

In this appendix we provide the missing proofs of theorems and claims in our main paper. We also describe detailed experimental settings here.

## A  TECHNICAL TOOLS

In this section we list the lemmas and theorems used during our proof.

**Lemma A.1** (Theorem 2.2, (Calabro, 2009)). Let $H_2^{-1}(s)$ be the inverse binary entropy function for $s \in [0, 1]$, then $H_2^{-1}(s) \geq s/2 \lg(6/s)$.

**Theorem A.1** (Data processing inequality). Let $X \perp Y \mid Z$, then $I(X; Z) \geq I(X; Y)$.

## B  MISSING PROOFS

**Proposition 2.1.** Let $h : \mathcal{X} \to \{0, 1\}$ be a hypothesis, then $\mathrm{Priv}_A(h) = 1$ iff $I(h(X); A) = 0$ and $\mathrm{Priv}_A(h) = 0$ iff $h(X) = A$ almost surely or $h(X) = 1 - A$ almost surely.

*Proof.* We first prove the first part of the proposition. By definition, $\mathrm{Priv}_A(h) = 1$ iff $\mathrm{Pr}_{\mathcal{D}}(h(X) = 1 \mid A = 1) = \mathrm{Pr}_{\mathcal{D}}(h(X) = 1 \mid A = 0)$, which is also equivalent to $h(X) \perp A$. It then follows that $h(X) \perp A \Leftrightarrow I(h(X); A) = 0$.

For the second part of the proposition, again, by definition of $\mathrm{Priv}_A(h)$, it is clear to see that we either have $\mathrm{Pr}_{\mathcal{D}}(h(X) = 1 \mid A = 1) = 1$ and $\mathrm{Pr}_{\mathcal{D}}(h(X) = 1 \mid A = 0) = 0$, or $\mathrm{Pr}_{\mathcal{D}}(h(X) = 1 \mid A = 1) = 0$ and $\mathrm{Pr}_{\mathcal{D}}(h(X) = 1 \mid A = 0) = 1$. Hence we discuss by these two cases. For ease of notation, we omit the subscript $\mathcal{D}$ from $\mathrm{Pr}_{\mathcal{D}}$ when it is obvious from the context which probability distribution we are referring to.

1. If $\mathrm{Pr}(h(X) = 1 \mid A = 1) = 1$ and $\mathrm{Pr}(h(X) = 1 \mid A = 0) = 0$, then we know that:

$$\begin{aligned}
\mathrm{Pr}(h(X) \neq A) &= \mathrm{Pr}(A = 0) \mathrm{Pr}(h(X) \neq A \mid A = 0) + \mathrm{Pr}(A = 1) \mathrm{Pr}(h(X) \neq A \mid A = 1) \\
&= \mathrm{Pr}(A = 0) \mathrm{Pr}(h(X) = 1 \mid A = 0) + \mathrm{Pr}(A = 1) \mathrm{Pr}(h(X) = 0 \mid A = 1) \\
&= \mathrm{Pr}(A = 0) \cdot 0 + \mathrm{Pr}(A = 1) \cdot 0 \\
&= 0.
\end{aligned}$$

2. If $\mathrm{Pr}(h(X) = 1 \mid A = 1) = 0$ and $\mathrm{Pr}(h(X) = 1 \mid A = 0) = 1$, similarly, we have:

$$\begin{aligned}
\mathrm{Pr}(h(X) \neq 1 - A) &= \mathrm{Pr}(A = 0) \mathrm{Pr}(h(X) \neq 1 - A \mid A = 0) + \mathrm{Pr}(A = 1) \mathrm{Pr}(h(X) \neq 1 - A \mid A = 1) \\
&= \mathrm{Pr}(A = 0) \mathrm{Pr}(h(X) = 0 \mid A = 0) + \mathrm{Pr}(A = 1) \mathrm{Pr}(h(X) = 1 \mid A = 1) \\
&= \mathrm{Pr}(A = 0) \cdot 0 + \mathrm{Pr}(A = 1) \cdot 0 \\
&= 0.
\end{aligned}$$

Combining the above two parts completes the proof. ∎

**Proposition 2.2.** If $\mathcal{H}$ is symmetric, then $\mathrm{Priv}_A(\mathcal{H}) = \min_{h \in \mathcal{H}} \mathrm{Pr}(h(X) = 0 \mid A = 1) + \mathrm{Pr}(h(X) = 1 \mid A = 0)$.

*Proof.* By definition, we have:

$$\begin{aligned}
\mathrm{Priv}_A(\mathcal{H}) &:= \min_{h \in \mathcal{H}} \mathrm{Priv}_A(h) \\
&= \min_{h \in \mathcal{H}} 1 - \big| \mathrm{Pr}(h(X) = 1 \mid A = 1) - \mathrm{Pr}(h(X) = 1 \mid A = 0) \big| \\
&= \min_{h \in \mathcal{H}} 1 - \big( \mathrm{Pr}(h(X) = 1 \mid A = 1) - \mathrm{Pr}(h(X) = 1 \mid A = 0) \big) \\
&= \min_{h \in \mathcal{H}} \mathrm{Pr}(h(X) = 0 \mid A = 1) + \mathrm{Pr}(h(X) = 1 \mid A = 0),
\end{aligned}$$

where the third equality holds due to the fact that $\max_{h \in \mathcal{H}} \big| \mathrm{Pr}(h(X) = 1 \mid A = 1) - \mathrm{Pr}(h(X) = 1 \mid A = 0) \big| = \max_{h \in \mathcal{H}} \big( \mathrm{Pr}(h(X) = 1 \mid A = 1) - \mathrm{Pr}(h(X) = 1 \mid A = 0) \big)$. To see this, for any specific $h$ such that the term inside the absolute value is negative, we can find $1 - h \in \mathcal{H}$ such that it becomes positive, due to the assumption that $\mathcal{H}$ is symmetric. ∎

**Proposition 3.1.** Let $f : \mathcal{X} \to \mathcal{Z}$ be a deterministic function and $\mathcal{H} \subseteq 2^{\mathcal{Z}}$ be a hypothesis class over $\mathcal{Z}$. For any joint distribution $\mathcal{D}$ over $X, A, Y$, if $I(f(X); A) = 0$, then $\mathrm{Priv}_A(\mathcal{H} \circ f) = 1$.

*Proof.* First, by the celebrated data-processing inequality, $\forall h \in \mathcal{H}$:

$$0 \leq I(h(f(X)); A) \leq I(f(X); A) = 0.$$

By Proposition 2.1, this means that $\forall h \in \mathcal{H}, \mathrm{Priv}_A(h) = 1$, which further implies that $\mathrm{Priv}_A(\mathcal{H} \circ f) = 1$ by definition. ∎

**Lemma 3.1.** For any feature map $f : \mathcal{X} \to \mathcal{Z}$, assume that $\mathcal{H}$ contains all the randomized binary classifiers, then o and $h^*(Z) := \arg\min_{h \in \mathcal{H}} \mathrm{CE}_Y(h \circ f) = \Pr(Y = 1 \mid Z = f(X))$.

*Proof.* Let $\mathcal{D}^f$ be the induced (pushforward) distribution of $\mathcal{D}$ under the map $f : \mathcal{X} \to \mathcal{Z}$. By the definition of cross-entropy loss, we have:

$$\begin{aligned}
\mathrm{CE}_Y(h \circ f) &= -\mathbb{E}_{\mathcal{D}} \left[ \mathbb{I}(Y = 0) \log(1 - h(f(X))) + \mathbb{I}(Y = 1) \log(h(f(X))) \right] \\
&= -\mathbb{E}_{\mathcal{D}^f} \left[ \mathbb{I}(Y = 0) \log(1 - h(Z)) + \mathbb{I}(Y = 1) \log(h(Z)) \right] \\
&= -\mathbb{E}_Z \mathbb{E}_{Y|Z} \left[ \mathbb{I}(Y = 0) \log(1 - h(Z)) + \mathbb{I}(Y = 1) \log(h(Z)) \right] \\
&= -\mathbb{E}_Z \left[ \Pr(Y = 0 \mid Z) \log(1 - h(Z)) + \Pr(Y = 1 \mid Z) \log(h(Z)) \right] \\
&= \mathbb{E}_Z \left[ D_{\mathrm{KL}}(\Pr(Y \mid Z) \mid\mid h(Z)) \right] + H(Y \mid Z) \\
&\geq H(Y \mid Z).
\end{aligned}$$

It is also clear from the above proof that the minimum value of the cross-entropy loss is achieved when $h(Z)$ equals the conditional probability $\Pr(Y = 1 \mid Z)$, i.e., $h^*(Z) = \Pr(Y = 1 \mid Z = f(X))$. ∎

**Theorem 3.1.** Let $f^*$ be the optimal feature map of (4) and define $H^* := H(A \mid Z = f^*(X))$. Then for any adversary $\widehat{A} : \mathcal{Z} \to \{0, 1\}$ such that $\widehat{A} \perp A \mid Z$, $\Pr_{\mathcal{D}^{f^*}}(\widehat{A} \neq A) \geq H^* / 2 \lg(6/H^*)$.

*Proof.* To prove this theorem, let $E$ be the binary random variable that takes value 1 iff $A \neq \widehat{A}$, i.e., $E = \mathbb{I}(A \neq \widehat{A})$. Now consider the joint entropy of $A, \widehat{A}$ and $E$. On one hand, we have:

$$H(A, \widehat{A}, E) = H(A, \widehat{A}) + H(E \mid A, \widehat{A}) = H(A, \widehat{A}) + 0 = H(A \mid \widehat{A}) + H(\widehat{A}).$$

Note that the second equation holds because $E$ is a deterministic function of $A$ and $\widehat{A}$, that is, once $A$ and $\widehat{A}$ are known, $E$ is also known, hence $H(E \mid A, \widehat{A}) = 0$. On the other hand, we can also decompose $H(A, \widehat{A}, E)$ as follows:

$$H(A, \widehat{A}, E) = H(E) + H(A \mid E) + H(\widehat{A} \mid A, E).$$

Combining the above two equalities yields

$$H(E) + H(A \mid E) + H(\widehat{A} \mid A, E) = H(A \mid \widehat{A}) + H(\widehat{A}).$$

Furthermore, since conditioning cannot increase entropy, we have $H(\widehat{A} \mid A, E) \leq H(\widehat{A})$, which further implies

$$H(A \mid \widehat{A}) \leq H(E) + H(A \mid E).$$

Now consider $H(A \mid E)$. Since $A \in \{0, 1\}$, by definition of the conditional entropy, we have:

$$H(A \mid E) = \Pr(E = 1)H(A \mid E = 1) + \Pr(E = 0)H(A \mid E = 0) = 0 + 0 = 0.$$

To lower bound $H(A \mid \widehat{A})$, realize that

$$I(A; \widehat{A}) + H(A \mid \widehat{A}) = H(A) = I(A; Z) + H(A \mid Z).$$

Since $\widehat{A}$ is a randomized function of $Z$ such that $A \perp \widehat{A} \mid Z$, due to the celebrated data-processing inequality, we have $I(A; \widehat{A}) \leq I(A; Z)$, which implies

$$H(A \mid \widehat{A}) \geq H(A \mid Z).$$

Combine everything above, we have the following chain of inequalities hold:

$$H(A \mid Z) \leq H(A \mid \widehat{A}) \leq H(E) + H(A \mid E) = H(E),$$

which implies

$$\Pr_{\mathcal{D}^{f*}} (A \neq \widehat{A}) = \Pr_{\mathcal{D}^{f*}} (E = 1) \geq H_2^{-1}(H(A \mid Z)),$$

where $H_2^{-1}(\cdot)$ is the inverse function of the binary entropy $H(t) := -t \log t - (1 - t) \log(1 - t)$ when $t \in [0, 1]$. To conclude the proof, we apply Lemma A.1 to further lower bound the inverse binary entropy function by

$$H_2^{-1}(H(A \mid Z)) \geq H(A \mid Z)/2 \lg(6/H(A \mid Z)),$$

completing the proof. ∎

**Lemma 4.2.** Let $\mathcal{D}_0$ and $\mathcal{D}_1$ be two distributions over $\mathcal{X}$ and let $\mathcal{D}_0^f$ and $\mathcal{D}_1^f$ be the induced distributions of $\mathcal{D}_0$ and $\mathcal{D}_1$ over $\mathcal{Z}$ by function $f$, then $d_{\mathrm{JS}}(\mathcal{D}_0^f, \mathcal{D}_1^f) \leq d_{\mathrm{JS}}(\mathcal{D}_0, \mathcal{D}_1)$.

*Proof.* Let $B$ be a uniform random variable taking value in $\{0, 1\}$ and let the random variable $Z_B$ with distribution $\mathcal{D}_B^f$ (resp. $X_B$ with distribution $\mathcal{D}_B$) be the mixture of $\mathcal{D}_0^f$ and $\mathcal{D}_1^f$ (resp. $\mathcal{D}_0$ and $\mathcal{D}_1$) according to $B$. It is easy to see that $\mathcal{D}_B = (\mathcal{D}_0 + \mathcal{D}_1)/2$, and we have:

$$
\begin{aligned}
I(B; X_B) &= H(X_B) - H(X_B \mid B) \\
&= -\sum \mathcal{D}_B \log \mathcal{D}_B + \frac{1}{2} \left( \sum \mathcal{D}_0 \log \mathcal{D}_0 + \sum \mathcal{D}_1 \log \mathcal{D}_1 \right) \\
&= -\frac{1}{2} \sum \mathcal{D}_0 \log \mathcal{D}_B - \frac{1}{2} \sum \mathcal{D}_1 \log \mathcal{D}_B + \frac{1}{2} \left( \sum \mathcal{D}_0 \log \mathcal{D}_0 + \sum \mathcal{D}_1 \log \mathcal{D}_1 \right) \\
&= \frac{1}{2} \sum \mathcal{D}_0 \log \frac{\mathcal{D}_0}{\mathcal{D}_B} + \frac{1}{2} \sum \mathcal{D}_1 \log \frac{\mathcal{D}_1}{\mathcal{D}_B} \\
&= \frac{1}{2} D_{\mathrm{KL}}(\mathcal{D}_0 \parallel \mathcal{D}_B) + \frac{1}{2} D_{\mathrm{KL}}(\mathcal{D}_1 \parallel \mathcal{D}_B) \\
&= D_{\mathrm{JS}}(\mathcal{D}_0, \mathcal{D}_1).
\end{aligned}
$$

Similarly, we have:

$$D_{\mathrm{JS}}(\mathcal{D}_0^f, \mathcal{D}_1^f) = I(B; Z_B).$$

Since $\mathcal{D}_0^f$ (resp. $\mathcal{D}_1^f$) is induced by $f$ from $\mathcal{D}_0$ (resp. $\mathcal{D}_1$), by linearity, $\mathcal{D}_B^f$ is also induced by $f$ from $\mathcal{D}_B$. Hence $Z_B = f(X_B)$ and the following Markov chain holds:

$$B \to X_B \to Z_B.$$

Apply the data processing inequality, we have

$$D_{\mathrm{JS}}(\mathcal{D}_0, \mathcal{D}_1) = I(B; X_B) \geq I(B; Z_B) = D_{\mathrm{JS}}(\mathcal{D}_0^f, \mathcal{D}_1^f).$$

Taking square root on both sides of the above inequality completes the proof. ∎

**Lemma 4.3.** Let $\hat{Y} = h(f(X)) \in \{0, 1\}$ be the predictor, then for $a \in \{0, 1\}$, $d_{\mathrm{JS}}(\mathcal{D}_a^Y, \mathcal{D}_a^{h \circ f}) \leq \sqrt{\mathrm{Err}_a(h \circ f)}$.

*Proof.* For $a \in \{0, 1\}$, by definition of the JS distance:

$$
\begin{aligned}
d_{\mathrm{JS}}^2(\mathcal{D}_a^Y, \mathcal{D}_a^{h \circ f}) &= D_{\mathrm{JS}}(\mathcal{D}_a^Y, \mathcal{D}_a^{h \circ f}) \\
&\leq \|\mathcal{D}_a^Y - \mathcal{D}_a^{h \circ f}\|_1/2 && \text{(Lemma 4.1)} \\
&= (|\Pr(Y = 0 \mid A = a) - \Pr(h(f(X)) = 0 \mid A = a)| \\
&\quad + |\Pr(Y = 1 \mid A = a) - \Pr(h(f(X)) = 1 \mid A = a)|)/2 \\
&= |\Pr(Y = 1 \mid A = a) - \Pr(h(f(X)) = 1 \mid A = a)| \\
&= |\mathbb{E}[Y \mid A = a] - \mathbb{E}[h(f(X)) \mid A = a]| \\
&\leq \mathbb{E}[|Y - h(f(X))| \mid A = a] \\
&= \mathrm{Err}_a(h \circ f),
\end{aligned}
$$

where the expectation is taken over the joint distribution of $X, Y$. Taking square root at both sides then completes the proof. ∎

**Theorem 4.1.** Let $\mathcal{H} \subseteq 2^{\mathcal{Z}}$ contains all the classifiers from $\mathcal{Z}$ to $\{0,1\}$. Given the conditions in Lemma 4.4, $\forall h \in \mathcal{H}$, $\mathrm{Util}_0(h \circ f) + \mathrm{Util}_1(h \circ f) + \mathrm{Priv}_A(\mathcal{H} \circ f) \leq 3 - \frac{1}{3} D_{\mathrm{JS}}(\mathcal{D}_0^Y, \mathcal{D}_1^Y)$.

*Proof.* Before we delve into the details, we first give a high-level sketch of the main idea. The proof could be basically partitioned into two parts. In the first part, we will show that when $\mathcal{H}$ contains all the measurable prediction functions, $1 - \mathrm{Priv}_A(\mathcal{H} \circ f)$ could be used to upper bound $D_{\mathrm{JS}}(\mathcal{D}_0^f, \mathcal{D}_1^f)$. The second part combines Lemma 4.3 and Lemma 4.2 to complete the proof.

In this part we first show that $D_{\mathrm{JS}}(\mathcal{D}_0^f, \mathcal{D}_1^f) \leq 1 - \mathrm{Priv}_A(\mathcal{H} \circ f)$:

$$D_{\mathrm{JS}}(\mathcal{D}_0^f, \mathcal{D}_1^f) \leq \frac{1}{2} \|\mathcal{D}_0^f - \mathcal{D}_1^f\|_1$$
$$= d_{\mathrm{TV}}(\mathcal{D}_0^f, \mathcal{D}_1^f)$$
$$= \sup_{A \in \mathscr{B}} |\mathcal{D}_0^f(A) - \mathcal{D}_1^f(A)|,$$

where $d_{\mathrm{TV}}(\cdot, \cdot)$ denotes the total variation distance and $\mathscr{B}$ is the sigma algebra that contains all the measurable subsets of $\mathcal{Z}$. On the other hand, when $\mathcal{H}$ contains all the measurable functions in $2^{\mathcal{Z}}$, we have:

$$1 - \mathrm{Priv}_A(\mathcal{H} \circ f) = 1 - \min_{h \in \mathcal{H}} \left(1 - |\Pr(h(Z) = 1 \mid A = 0) - \Pr(h(Z) = 1 \mid A = 1)|\right)$$
$$= \max_{h \in \mathcal{H}} |\Pr(h(Z) = 1 \mid A = 0) - \Pr(h(Z) = 1 \mid A = 1)|$$
$$= \max_{h \in \mathcal{H}} |\mathcal{D}_0(h^{-1}(1)) - \mathcal{D}_1(h^{-1}(1))|$$
$$= \sup_{A \in \mathscr{B}} |\mathcal{D}_0(A) - \mathcal{D}_1(A)|,$$

where the last equality follows from the fact that $\mathcal{H}$ is complete and contains all the measurable functions. Combine the above two parts we immediately have $D_{\mathrm{JS}}(\mathcal{D}_0^f, \mathcal{D}_1^f) \leq 1 - \mathrm{Priv}_A(\mathcal{H} \circ f)$.

Now using the key lemma, we have:

$$d_{\mathrm{JS}}(\mathcal{D}_0^Y, \mathcal{D}_1^Y) \leq d_{\mathrm{JS}}(\mathcal{D}_0^Y, \mathcal{D}_0^{h \circ f}) + d_{\mathrm{JS}}(\mathcal{D}_0^f, \mathcal{D}_1^f) + d_{\mathrm{JS}}(\mathcal{D}_1^{h \circ f}, \mathcal{D}_1^Y)$$
$$\leq \sqrt{\mathrm{Err}_0(h \circ f)} + \sqrt{1 - \mathrm{Priv}_A(\mathcal{H} \circ f)} + \sqrt{\mathrm{Err}_1(h \circ f)}$$
$$= \sqrt{1 - \mathrm{Util}_0(h \circ f)} + \sqrt{1 - \mathrm{Priv}_A(\mathcal{H} \circ f)} + \sqrt{1 - \mathrm{Util}_1(h \circ f)}$$
$$\leq \sqrt{3(1 - \mathrm{Util}_0(h \circ f) + 1 - \mathrm{Util}_1(h \circ f) + 1 - \mathrm{Priv}_A(\mathcal{H} \circ f))}$$
$$= \sqrt{3\big(3 - (\mathrm{Util}_0(h \circ f) + \mathrm{Util}_1(h \circ f) + \mathrm{Priv}_A(\mathcal{H} \circ f))\big)}.$$

Taking square at both sides and then rearrange the terms then completes the proof. ∎

**Theorem 4.2.** Assume the conditions in Theorem 4.1 hold. If $\mathrm{Priv}_A(\mathcal{H} \circ f) \geq 1 - D_{\mathrm{JS}}(\mathcal{D}_0^Y, \mathcal{D}_1^Y)$, then $\forall h \in \mathcal{H}$, $\mathrm{Err}_0(h \circ f) + \mathrm{Err}_1(h \circ f) \geq \frac{1}{2}\big(d_{\mathrm{JS}}(\mathcal{D}_0^Y, \mathcal{D}_1^Y) - \sqrt{1 - \mathrm{Priv}_A(\mathcal{H} \circ f)}\big)^2$.

*Proof.* Similarly, using the key lemma, we have:

$$d_{\mathrm{JS}}(\mathcal{D}_0^Y, \mathcal{D}_1^Y) \leq d_{\mathrm{JS}}(\mathcal{D}_0^Y, \mathcal{D}_0^{h \circ f}) + d_{\mathrm{JS}}(\mathcal{D}_0, \mathcal{D}_1) + d_{\mathrm{JS}}(\mathcal{D}_1^{h \circ f}, \mathcal{D}_1^Y)$$
$$\leq \sqrt{\mathrm{Err}_0(h \circ f)} + \sqrt{1 - \mathrm{Priv}_A(\mathcal{H} \circ f)} + \sqrt{\mathrm{Err}_1(h \circ f)}$$

Under the assumption that $\mathrm{Priv}_A(\mathcal{H} \circ f) \geq 1 - D_{\mathrm{JS}}(\mathcal{D}_0^Y, \mathcal{D}_1^Y)$, we have $d_{\mathrm{JS}}(\mathcal{D}_0^Y, \mathcal{D}_1^Y) \geq \sqrt{1 - \mathrm{Priv}_A(\mathcal{H} \circ f)}$, hence by AM-GM inequality:

$$\sqrt{2\big(\mathrm{Err}_0(h \circ f) + \mathrm{Err}_1(h \circ f)\big)} \geq \sqrt{\mathrm{Err}_0(h \circ f) + \mathrm{Err}_1(h \circ f)} \geq d_{\mathrm{JS}}(\mathcal{D}_0^Y, \mathcal{D}_1^Y) - \sqrt{1 - \mathrm{Priv}_A(\mathcal{H} \circ f)}.$$

Taking square at both sides then completes the proof. ∎

## C    Multi-attribute Defense

Although our discussion in the paper only focuses on the case where there is only one sensitive attribute that the data vendor would like to protect, our optimization framework is flexible enough to extend to the setting where multiple sensitive attributes need to be preserved simultaneously. For instance, in online advertising, the data vendor often needs to keep the personal information about specific users secret, e.g., age range, demographic group, income level, etc. To this end, let $\{A_i\}_{i=1}^K$ be $K$ sensitive attributes that the data vendor would like to protect.

Define $\varepsilon_i := \Pr_{\mathbf{S}}(h_i'(f(X)) = 0 \mid A_i = 1) + \Pr_{\mathbf{S}}(h_i'(f(X)) = 1 \mid A_i = 0)$ to simplify the notation. Similar to the optimization formulation in (1), the following general optimization formulation handles the setting of multi-attribute defense:

$$\textbf{Hard version:} \qquad \min_{h \in \mathcal{H}, f} \max_{h_1', \ldots, h_K' \in \mathcal{H}} \quad \widehat{\mathrm{Err}}(h \circ f) + \lambda \max_{i \in [K]}(-\varepsilon_i) \qquad (6)$$

Problem (6) is still a minimax optimization problem. If we initialize all the functions, including $f, h$ and $\{h_i'\}_{i=1}^K$ using deep neural networks, then (6) turns into a nonconvex minimax optimization problem. Inspired by Ganin et al. (2016), we can use the gradient reversal layer to effectively implement (6) by backpropagation. Essentially, with the gradient reversal layer, we use the Gradient Descent/Ascent (GDA) (Daskalakis & Panageas, 2018) algorithm to optimize all the model parameters, as opposed to the alternative gradient algorithm, which is known to be unstable in the nonconvex setting (Goodfellow et al., 2014).

One notable drawback of (6) is that in each iteration, the forward evaluation phase requires computation over all the $K$ adversaries, while in the backward propagation phase only one of them is being utilized due to the hard $\max$ operator. This is rather data-inefficient and can waste our computational resources in the forward evaluation phase. To avoid this problem, we propose a smoothed formulation of (6) using the fact that $\frac{1}{\gamma} \log \sum_{i \in [K]} \exp(-\gamma \varepsilon_i) \to \max_{i \in [K]}(-\varepsilon_i)$ as $\gamma \to \infty$:

$$\textbf{Smooth version:} \qquad \min_{h \in \mathcal{H}, f} \max_{h_1', \ldots, h_K' \in \mathcal{H}} \quad \widehat{\mathrm{Err}}(h \circ f) + \frac{\lambda}{\gamma} \log \sum_{i \in [K]} \exp(-\gamma \varepsilon_i) \qquad (7)$$

We call the one in (6) as hard version multi-attribute defense and (7) as the smooth version multi-attribute defense. Let $\theta$ denote the model parameters of $f$. Take the derivative w.r.t. $\theta$, we have:

$$\frac{\partial}{\partial \theta} \frac{1}{\gamma} \log \sum_{i \in [K]} \exp(-\gamma \varepsilon_i) = - \sum_{i \in [K]} \frac{\exp(-\gamma \varepsilon_i)}{\sum_{j \in [K]} \exp(-\gamma \varepsilon_j)} \frac{\partial \varepsilon_i}{\partial \theta}.$$

Compared with the hard version, the smooth version not only avoids the data-inefficiency problem, but also provides an adaptive way to combine the feedback from all the $K$ adversaries by convex combination. Intuitively, the above formulation suggests that during optimization, the larger the error from one adversary, the smaller the combination weight in the ensemble. This is consistent with Proposition 2.2 where we can see that a larger error essentially corresponds to a better protection of the corresponding sensitive attribute, hence a smaller combination weight.

To demonstrate that the effectiveness of multi-attribute protection, we also evaluate both the hard and smooth versions of our multi-attribute defense on UTKFace dataset. We compare both versions with no defense for both attributes and defenses for single attribute, as none of the other competitors has a multi-attribute defense extension. The results are shown in Figure 4.

In Figure 4, we can see that both the hard and smooth variants help to protect two private attributes, indicated by the low private accuracies in both cases. Notably, the target accuracy does not degrade too much as compared to the one in single-attribute defense. Among the two variants, the smooth variant is slightly more effective, possibly due to the adaptive combination property.

## D    Detailed Experiments

In this section, we provide more details of the experiments. First we provide the details of different existing methods we evaluate. Then we elaborate more dataset description, model architecture and training parameters in different experiments.

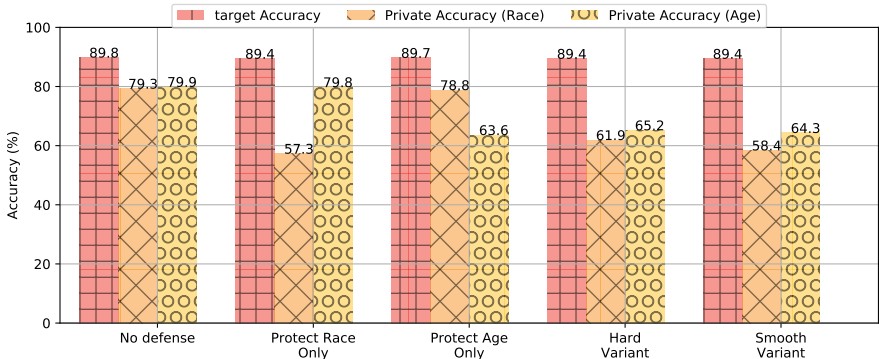

Figure 4: Classification accuracy of multi-attribute defense on the UTKFace dataset. For target accuracy, the larger accuracy the better. For private accuracy, the smaller accuracy the better.

## D.1 DETAILS ON METHODS

We provide a detailed description of each method here:

1). Privacy Partial Least Squares (PPLS): It learns $n \times X_d$ matrix for the feature transformation. The matrix is learned by maximizing the covariance of the learned representation and target attribute while minimizing the covariance of the learned representation and private attribute.

2). Privacy Linear Discriminant Analysis (PLDA): It learns $n \times X_d$ matrix for the feature transformation. The matrix is learned by maximizing the Fisher's linear discriminability of the learned representation and target attribute while minimizing the Fisher's linear discriminability of the learned representation and private attribute.

3). Minimax filter with alternative update (ALT-UP): The representation is learn via optimizing Equation 3 in an alternative way, first we update the parameters of the feature transformation module and the target attribute classifier, and then accordingly update the private attribute classifier.

4). Maximum Entropy Adversarial Representation Learning (MAX-ENT) (Roy & Boddeti, 2019): he objective equation is the slightly different from ALT-UP. The latter term contains additional entropy term to maximize unpredictability of the private attribute.

5). Gradient Reversal Layer (GRL): The objective equation is the same as ALT-UP, and we train the feature transformation module by adding a gradient reversal layer between the feature transformation module and the private attribute classifier.

6). Principal Component Analysis (PCA): It generates a $n \times X_d$ matrix for the feature transformation where the rows of the matrix are the $n$ largest eigenvectors of the input dataset $X$.

7). No defense (NO-DEF): It is equivalent to normal training by setting $\lambda = 0$ in Equation 3.

8). Local Differential Privacy (LDP): Standard Laplace mechanism of local differential privacy, where the noise is added to the raw representation for erasing the information of the private attribute.

9). Differentially private SGD (DPSGD) (Abadi et al., 2016): It is one of the state-of-the-art differential privacy methods on deep learning. It adds Gaussian noise to the gradients when training the model.

## D.2 DETAILS ON UCI ADULT DATASET EVALUATION

UCI Adult dataset is a benchmark machine learning dataset for income prediction. Each data record contains 14 categorical or numerical attributes, such as occupation, education and gender, to predict whether individual annual income exceeds $50K/year. The dataset is divided into training set (24130 examples), validation (6032 examples), and test set (15060 examples). We choose gender, age, and education as the private attributes, respectively.

Table 2: Data distribution of income ($Y$) and gender ($A$) in UCI Adult dataset.

|  | $Y = 0$ | $Y = 1$ |
|---|---|---|
| $A = 0$ | 20988 | 9539 |
| $A = 1$ | 13026 | 1669 |

Table 3: Data distribution of income ($Y$) and age ($A$) in UCI Adult dataset.

|  | $Y = 0$ | $Y = 1$ |
|---|---|---|
| $A = 0$ | 18042 | 2473 |
| $A = 1$ | 15972 | 8735 |

Table 4: Data distribution of income ($Y$) and education ($A$) in UCI Adult dataset.

|  | $Y = 0$ | $Y = 1$ |
|---|---|---|
| $A = 0$ | 20447 | 4248 |
| $A = 1$ | 13567 | 6960 |

We process each private attribute as binary label for each experiment: for age label, 0 if the person is no greater than 35 years old and 1 otherwise; for education label, 0 if the person has not entered college or receive higher education than college, and 1 otherwise. In the mean time, we also remove corresponding private attribute from the input, so the dimension of input data for each experiment is different. The input dimensions for income-gender experiment, income-age experiment, and income-education experiment are 113, 104 and 99, respectively. Table 2, Table 3 and Table 4 summarize the data distribution of UCI Adult dataset for protecting different private attributes.

We use the two-layer ReLU-based neural net for $f$ and one-layer neural net for $h$. The output dimensions of $f$ are 64. We train all methods using SGD with the initial learning late 0.001 and momentum 0.9 for 40 epochs. In the DP-SGD experiment, we set the noise multiplier as 0.45 and 4.0 for small noise and large noise, respectively, and set the clipping norm as 1.0. $(\epsilon, \delta)$ for DPSGD small noise and DPSGD large noise are $(33.7, 10^{-5})$ and $(0.572, 10^{-5})$, respectively. Among all methods, we report the one achieving the best performance on the target task in the validation set. We run the experiments for ten times and compute the average.

### D.3    DETAILS ON UTKFACE DATASET EVALUATION

UTKFace dataset is a large scale face dataset with annotations of age (range from 0 to 116 years old), gender (male and female), and ethnicity (White, Black, Asian, Indian, and Others). It contains 23,705 $64 \times 64$ aligned and cropped RGB face images and we split the dataset into training set (15171 examples), validation set (3793 examples) and test set (4741 examples), respectively. We further process age label and ethnicity label as binary labels: 0 if the person is not greater than 35 years old for age label (is white for ethnicity label), and 1 if the the person is greater than 35 years old for age label (is non-white for ethnicity label). Table 5 and Table 6 summarize the data distribution of UTKFace dataset for protecting different private attributes.

Table 5: Data distribution of gender ($Y$) and race ($A$) in UTKFace dataset.

|  | $Y = 0$ | $Y = 1$ |
|---|---|---|
| $A = 0$ | 5477 | 4601 |
| $A = 1$ | 6914 | 6713 |

Table 6: Data distribution of gender ($Y$) and age ($A$) in UTKFace dataset.

|  | $Y = 0$ | $Y = 1$ |
|---|---|---|
| $A = 0$ | 6889 | 8218 |
| $A = 1$ | 5502 | 3096 |

Since NO-DEF, ALT-UP, GRL and DP can directly enjoy the benefits of using the state-of-the-art neural network architecture as feature extraction module, so we use the feature extraction module of Wide Residual Network (Zagoruyko & Komodakis, 2016) for the (non-linear) feature transformation module, while PPLS, PLDA, and PCA learn $12288 \times 2048$ matrix filter for $f$. We train all methods using SGD with the initial learning late 0.01 and momentum 0.9 for 50 epochs. The learning rate is decayed by a factor of 0.1 for every 20 epochs. In the DP-SGD experiment, we set the noise

multiplier as 0.45 and 1.0 for small noise and large noise, respectively, and set the clipping norm as 1.0. $(\epsilon, \delta)$ for DPSGD small noise and DPSGD large noise are $(25.7, 10^{-5})$ and $(2.7, 10^{-5})$, respectively. Among all methods, we report the one achieving the best performance on the target task in the validation set. We run the experiments for ten times and compute the average.

For the experiment of multi-attribute defense, we choose $\lambda$ to be 3 for both hard and smooth variants. We find that this is a reasonable choice of $\lambda$ since $\lambda$ cannot be too large (otherwise it will cause gradient explosion during training) and cannot be too small (otherwise the private task accuracy is still too high) in this learning task. All other parameter settings are the same as the ones described before.

