# OpenReview forum: "Adversarial Privacy Preservation under Attribute Inference Attack"
_ICLR.cc/2020/Conference — Reject_

### Official Review · AnonReviewer3 · 2019-10-09
**Official Blind Review #3**

**Rating:** 3

**Review:**

Summary
In this paper, the author introduces a new privacy notation for the attribute attacks. Under the notation, the author has theoretically analyzed the trade-off between privacy preservation and model utility. Based on the theoretical finding,
they further propose an adversarial representation learning paradigm to achieve the best trade-off.

Strengths:
1. This paper provides an interesting information-theoretic view to study the privacy-preserving machine learning algorithm.
Based on this view, the author has presented a comprehensive analysis of the trade-off between privacy-preserve and model
utility.
2. The topic of studying the privacy guarantee against attribute attacks is important and implies a wide range of applications, such as preventing the model-inversion attack.
3. The paper is well-written and provides an enjoyable reading experience.

Weakness:
1. In experiments, the DP based method should be an important comparison method. However, the DP method used in this paper
seems to be a weak baseline, which injects the noise into the raw data. Many prior works have prooved that injecting noise
into the gradient leads to a better trade-off between the utility and the privacy budget [1]. Thus, the author should re-design their DP baseline and provides comparison results.
2. There is a related work [2] of reducing the privacy leakage of the feature representation, which also takes a view from the information-theoretic view (i.e. a maximum entropy approach). Although this work focuses on the task of image representation, the author also conducts experiments on a non-image dataset, i.e, UCI dataset. The comparative experiments need to be conducted to show the effectiveness of the proposed method.
3. The notation of the distribution is confusing. It seems that the author referred $\mathcal{D}$  to both three joint distributions?

[1]  Abadi et al. Deep Learning with Differential Privacy
[2] Mitigating Information Leakage in Image Representations: A Maximum Entropy Approach

**Experience Assessment:**

I have published one or two papers in this area.

**Review Assessment: Checking Correctness Of Derivations And Theory:**

I assessed the sensibility of the derivations and theory.

**Review Assessment: Checking Correctness Of Experiments:**

I carefully checked the experiments.

**Review Assessment: Thoroughness In Paper Reading:**

I read the paper at least twice and used my best judgement in assessing the paper.

---

> ### Author Response · Authors · 2019-11-09
> **Response**
>
> We thank the reviewer for the thoughtful comments, and we are happy that the reviewer find our information-theoretic view important and interesting. In what follows we attempt to answer reviewer's questions.
>
> # More baselines
> Thanks for the suggestions. In the updated manuscript we perform additional experiments on the two datasets with both [1] and [2]. The updated results are now available in the updated version of the manuscript.
>
> We refer the existing DP mechanism used in the manuscript as local DP and the added baseline as DP-SGD. In summary, local DP is a stronger notion than DP-SGD and it helps to preserve privacy at the cost of greater utility loss. On the other hand, as we commented in our response to Reviewer 2, the existing definition of DP does not prevent attribute inference attack, and we confirmed this from our experiments, which show that DP-SGD fails to protect against attribute inference attack on the two datasets. However, DP-SGD is better than local DP in terms of preserving utility.
>
> For the MaxEnt approach, we found that the overall results are similar to other adversarial representation learning methods when $\lambda$ is small. However, MaxEnt is more unstable compared to other adversarial representation methods such as GRL when $\lambda$ is large. Detailed results are shown in Figure 1 and 2. In summary, the MaxEnt method is more sensitive to the choice of $\lambda$. The results are reasonable since additional entropy term is incorporated in the objective function and a large $\lambda$ will increase uncertainty during training.
>
>
> [1].    Deep Learning with Differential Privacy, Abadi et al., CCS 2016.
> [2].    Mitigating Information Leakage in Image Representations: A Maximum Entropy Approach, Roy et al., CVPR 2019.
>
> # Notation
> We use $\mathcal{D}$ to refer the joint distribution over $A, X$ and $Y$.

---

### Official Review · AnonReviewer2 · 2019-10-23
**Official Blind Review #2**

**Rating:** 6

**Review:**

This paper proposes a definition of privacy against attribute inference attack and formulates it as a minimax optimization problem that balances accuracy and privacy, it then provides an information-theoretic lower bound for the inference error. It also provides an analysis of the intrinsic privacy-utility tradeoff. Finally, it shows experimental evaluation in terms of inference error and accuracy of several representation-learning based approaches for protecting against attribute inference attack and compares with the lower bounded proved in the paper. By comparing utility with a differentially private algorithm, it shows that the other algorithm achieves higher utility under similar privacy.

The formulation of the minimax problem seems pretty interesting and the information-theoretic bound seems pretty useful in measuring inference risk based on the experimental results.
As for the comments and experiments with differential privacy, I’m not sure if I follow the argument correctly. It seems to me DP is stronger than the privacy guarantee defined in the paper, as it prevents an adversary from knowing whether record x or x’ is in the dataset, which I think directly implies that an adversary won’t be able to infer whether a record with attribute A or A’ (with the rest of the attributes kept) is in the dataset. Can you comment more on that side? Besides, there were two epsilon values used in the experiments for DP. As your goal is not to guarantee DP anyway, I think it makes sense to try more epsilon values, especially much larger epsilon values, as the analyzed epsilon is sometimes pretty loose upper bound.

**Experience Assessment:**

I do not know much about this area.

**Review Assessment: Checking Correctness Of Derivations And Theory:**

I assessed the sensibility of the derivations and theory.

**Review Assessment: Checking Correctness Of Experiments:**

I assessed the sensibility of the experiments.

**Review Assessment: Thoroughness In Paper Reading:**

I read the paper at least twice and used my best judgement in assessing the paper.

---

> ### Author Response · Authors · 2019-11-09
> **Response**
>
> We thank the reviewer for providing the positive comments. We attempt to answer the reviewer's question below.
>
> # DP
> Yes, by definition, standard notion of DP provides privacy against adversary that tries to guess whether one specific instance appears in the training data or not. Formally, this problem is more related to the membership inference attack problem in the literature [1]. Intuitively, one can understand a given data set as a 2D matrix where the row of the matrix index the membership and the column of the matrix index the attribute. From this perspective, the goal of membership inference attack is to guess the row while attribute inference attack is to guess the column, hence these two are not the same. While by definition DP helps to directly protect against membership inference attack, it does not necessarily help to protect against attribute inference attack. Put in other words, DP guarantees that the output of a randomized algorithm (in our case the output is the parameters of the neural nets) are similar with two adjacent datasets, but the model itself can still contain information related to a particular attribute $A$ [2].
>
> More detailed discussion on the connection and difference between these two topics could be found in the "Algorithmic Fairness" paragraph of the Related Work section and Figure 3 in our manuscript.
>
>
> # More $\epsilon$ values in DP
> Thanks for the suggestions. We've added more experiments with DP using larger $\epsilon$ values. The new results are still consistent with the existing ones in terms of privacy-utility tradeoff. Please check the updated manuscript for a more detailed description.
>
> [1].    Membership Inference Attacks Against Machine Learning Models, Shokri et al., S&P, 2017.
> [2]. Property inference attacks on fully connected neural networks using permutation invariant representations, Ganju, Karan, et al., in CCS 18

---

### Official Review · AnonReviewer1 · 2019-10-24
**Official Blind Review #1**

**Rating:** 3

**Review:**

Overall, this paper provides valuable insight into the trade-off between privacy preservation and utility when training a representation to fight against attribute inference attack.

Detail comments:

Strength:
+ The whole paper is well organized with logic. Notations are well defined and distinguished.
+ The final results have a good intuitive explanation.
+ The most impressive part of this paper is the analysis of the trade-off between privacy and utility from which the upper bound is quantified.

Weakness:
+ The minimax method looks trivial. The difficulty of using such an objective should be emphasized for practical implementation.
+ The major weakness is the experiments. The experiments only on two datasets may not be convincing for me. And the repetition times, 5 or 3, for each dataset are pretty small. Considering the experiments are conducted with random noise, e.g., DP methods, such a small repetition time is not fair since there a large chance these results could be selected.
+ Which DP Laplacian mechanism is used is not specified. Since there are already many improvements on the DP Laplacian mechanism, e.g., [A], it is necessary to make sure the baseline should be state-of-the-art for a fair comparison.
+ The result is not intuitively surprising that the privacy loss and utility loss will be balanced toward an upper bound (Theorem 4.2). I am not sure how the Jensen-Shannon entropy between D^Y_0 and D^Y_1 can be calculated in practice since the true conditional distribution is not observed. For example, when the data distribution is heavily biased, then the conditional distribution might show less correlation between Y and A. Then, the privacy protection will be a pretty simple task with a very high upper bound. Based on this, it is worth to ask what the upper bound looks like in the dataset used in the experiments.
+ How efficient this algorithm could be? In comparison to other baselines, does this method provide a more efficient solution?

[A] Phan, N., Wu, X., Hu, H., & Dou, D. (2017). Adaptive Laplace Mechanism: Differential Privacy Preservation in Deep Learning. 2017 IEEE International Conference on Data Mining (ICDM), 385–394. https://doi.org/10.1109/ICDM.2017.48

**Experience Assessment:**

I have published one or two papers in this area.

**Review Assessment: Checking Correctness Of Derivations And Theory:**

I did not assess the derivations or theory.

**Review Assessment: Checking Correctness Of Experiments:**

I carefully checked the experiments.

**Review Assessment: Thoroughness In Paper Reading:**

I read the paper at least twice and used my best judgement in assessing the paper.

---

> ### Author Response · Authors · 2019-11-09
> **Response**
>
> We thank the reviewer for providing thoughtful comments, and we appreciate that the reviewer find our theoretical results to have valuable insights. We answer the reviewer's question below.
>
> # The minimax method
> We described the optimization algorithm we used in practice to optimize the minimax objective function in the third paragraph of Appendix C. In a nutshell, we use the gradient reversal technique to implement the gradient descent/ascent algorithm, which has been recently shown [1] to generate limit points that are stable and are local min-max.
>
> # Experiments
> As suggested by the reviewer, we added more experiments by repeating each algorithm on each dataset for 10 times and report both the mean and the standard deviation of the final results. The updated results are now available in the updated version of the manuscript. As a summary, overall we observe the same phenomenon as our original results: DP usually helps to preserve privacy, but at the great cost of utility loss. On the other hand, representation learning based algorithms usually achieve better privacy-utility trade-offs. More detailed discussion about the experiments could be found in Section 5.2 of the updated manuscript.
>
> # DP algorithm
> In the updated manuscript, besides the local DP mechanism used in the previous version, we also add new experiments to compare with the state-of-the-art DP-SGD method [2] that is specifically designed and tailored for learning with deep neural networks. The results could be found in Fig.1. As a summary, the DP-SGD fails to protect against attribute inference attack as it is only designed to protect membership inference attack (See more discussion on this point in our response to Reviewer 2). However, DP-SGD is better than local DP in terms of utility loss, but the utility loss is still greater than adversarial representation learning in most cases. We also cited and briefly discussed [4] in the update related work section.
>
> # Computation of JS distance
> Yes, in practice it's both computationally and statistically intractable to estimate the JS distance. The reason is that for high-dimensional continuous distribution, the sample complexity of estimating information-theoretic quantities, including mutual information, Shannon entropy, KL divergence, etc., is exponential in the number of dimensions. That being said, one alternative way to do so is to consider the variational representations of KL-divergence and use rich parameterized function class (e.g., neural networks) to approximate these distances. For example, recent work [3] on estimating mutual information has empirically shown that such an approach often leads to better estimation result than classic approaches based on nonparametric density estimation.
>
> # Efficiency of the algorithm
> All the algorithms compared in the experiments scale linearly in the size of the dataset. For representation learning based methods, since we use the same base networks for inference, the running time are also comparable between them.
>
> [1].    The Limit Points of (Optimistic) Gradient Descent in Min-Max Optimization, Daskalakis et al., NeurIPS 2018.
> [2].    Deep Learning with Differential Privacy, Abadi et al., CCS 2016.
> [3].    MINE: Mutual Information Neural Estimation. Belghazi et al., ICML 2018.
> [4].   Adaptive Laplace Mechanism: Differential Privacy Preservation in Deep Learning. Phan et al., ICDM 2017.
>
> We hope our response has answered reviewer's questions and helped clarify our description of the work.

---

### Author Response · Authors · 2019-11-09
**General Response**

We thank all the reviewers for the thoughtful comments and we answer each reviewer’s questions individually below. We have also performed additional experiments as requested by Reviewer 1 and 3, and updated our manuscript accordingly. More detailed results and discussions of the additional experiments could be found in Section 5.

---

### Decision · Program_Chairs · 2019-12-19

**Decision:**

Reject

**Comment:**

While there was some support for the ideas presented, the majority of reviewers felt that this submission is not ready for publication at ICLR in its present form.

The most significant concerns raised were about the strength of the experiments, and choice of appropriate baselines.

---

> ### Author Response · Authors · 2019-12-20
> **Concerns were addressed in our updated version**
>
> We thank the meta-reviewer for the summary. However, we would like to bring it to the attention of the reviewers as well as the meta-reviewer that we had already addressed the main concerns in our updated version of the paper by 1). Adding more baseline experiments using the mentioned DP algorithms 2). Strengthening the existing experiments by repeating more times to reduce the variance of the randomness. We hope that the reviewers and the meta-reviewer could have a check of our updated paper.